# Energy Communities in Urban Areas: Comparison of Energy Strategy and Economic Feasibility in Italy and Spain

Simona Barbaro * and Grazia Napoli

Department of Architecture, University of Palermo, 90133 Palermo, Italy; grazia.napoli@unipa.it
* Correspondence: simona.barbaro@unipa.it

**Abstract:** Energy communities using renewable energy sources directly contributes to reduction of climate change gas emissions and energy consumption in the European Union. In addition, energy communities enable citizens to transform from (passive) consumers to prosumers (active consumers and producers) and to play a proactive role in the deployment of energy transition in urban areas. As the transposition of European rules about energy communities into the national laws of EU Member States is very articulated and differentiated, this study proposes a framework to analyze and compare regulatory and financial instruments. This framework is applied to the analysis of the cases of Italy and Spain as representative of European states in which collective actions in the energy sector are not very common, with the aim of highlighting the main critical issues affecting the effectiveness of energy transition strategies and assessing the economic feasibility of energy communities. Based on analysis of regulations and procedures, including at the local level, it appears that municipalities play an important role as promoters of initiatives among citizen communities, while complex bureaucratic procedure is the most critical issue in both countries and can significantly hinder the spread of energy communities. With respect to the different financial incentives available for the formation of energy communities in Italy and Spain, a few cases studies are hypothesized, calculating the most relevant cost-effectiveness indicators, e.g., Net Present Value. It turns out that a project with the same characteristics achieves greater economic feasibility in Italy than in Spain, depending on the type and size of incentives set by national laws and, above all, that financial incentives are necessary to make the formation of energy communities cost-effective and thus to achieve direct citizen involvement in energy transition actions.

**Keywords:** energy communities; financial incentives; economic feasibility





## 1. Introduction

Environmental and energy policies of European Union (EU) actively promote the transition to a low-carbon society and sustainable energy systems and to reach these goals the EU signed the Paris Agreement on climate change [1] and adopted the 17 Sustainable Development Goals (SDGs) in the 2030 Agenda for Sustainable Development [2] (Figure 1).

The SDGs promoted by the United Nations (UN) in 2015 can only be achieved if there is a radical change in the management of natural resources, which includes drastically lowering greenhouse gases emissions as well as reorganizing economic systems according to the principles of circular economy. For this reason, the European Commission (EC) has progressively committed itself to issuing policies and directives and implementing instruments to support environmental protection and sustainable development in the European Union. Several official acts, such as "A policy framework for climate and energy in the period from 2020 to 2030" or "The European Green Deal" [3–5], involve energy efficiency and retrofitting actions on the urban and building scales. In fact, transforming urban energy systems is one of the key factors in ensuring accessible, green and secure energy services for all and in making cities and communities inclusive, sustainable and

resilient, as required by "Goal 7: Clean and Affordable Energy" and "Goal 11: Sustainable Cities and Communities" of the SDGs, respectively.

**Figure 1.** The 17 Sustainable Development Goals (SDGs) (source: United Nations).

SDGs 7 and 11 therefore recognize the close interrelationships between the city and the energy sector to ensure the welfare of citizens, reduce social inequalities and preserve the natural environment while also reducing the effects of climate change [6]. Cities are required to take an active part in this renewal process and have to adopt new patterns of urban space and public and private mobility, waste management and supply of natural resources, especially by imposing high standards of energy efficiency on urban building stock [7]. In addition, on the urban and building scales, both a drastic reduction in energy demand for the air conditioning of buildings through the implementation of energy efficiency measures and the territorial spread of renewable energy sources are required to accelerate energy transition.

The foresting of strategies and actions to make urban environments and buildings more sustainable, both in the private and public sectors, is a cornerstone of the European Union's long-term vision for a climate-neutral economy. Economic and financial feasibility analysis of energy efficiency measures on the building, district and city scale have been widely applied to the evaluation of energy retrofit projects for near-zero energy buildings (NZEB) and various types of existing buildings [8–11], as well as for the definition of best urban energy scenarios [12,13], also in the presence of public incentives and supporting public decisions with multi-criteria models [14,15].

In 2010, the achievement of energy neutrality at the building scale had already been tested in Net Zero Energy Buildings (NZEB) and, similarly, energy neutrality on the district scale had been applied in Net Zero Energy Districts (NZED), where, in addition to energy efficiency measures in buildings, renewable energy production systems capable of achieving high levels of efficiency due to the economies of scale associated with the size of the district as opposed to the size of the individual building have been explored [16]. To this end, the European Union launched the Renovation Wave Strategy in October 2020 as part of the European Green Deal [17], which aims to double renovation rates over the next ten years to reduce energy consumption in buildings as well as to increase financing opportunities to support the implementation of energy retrofitting projects.

The results of these experiments have shown, however, that the economic feasibility of retrofit measures is often not attained without public subsidies [18,19]. On the other hand, public incentives have an enormous influence on the selection and implementation of

energy retrofit measures both in terms of ranking alternatives and validating environmental policies [20,21].

Energy efficiency measures in buildings aim to decrease final energy consumption but, obviously, do not reduce it to zero; therefore, to further lower greenhouse gas emissions, it is necessary to accelerate energy transition locally and globally through the use of renewable energy sources (RESs). According to Eurostat data [22], the share of energy from renewable sources in EU countries has been gradually growing from an average of 9.6% in 2004 to 21.8% in 2021 (Figure 2), but it must increase significantly in the years ahead to meet the new mandatory RES target of 42.5% by 2030 set in the 2023 revision of Directive 2018/2001/EU [23,24].

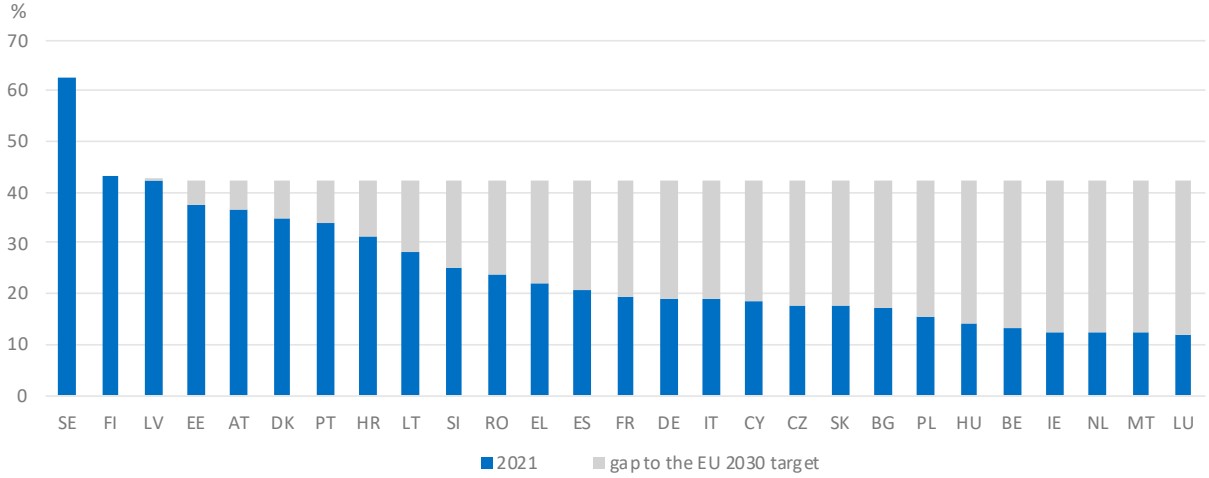

**Figure 2.** Share of renewable energy source in gross final energy consumption per EU member states in 2021 in relation to the EU 2030 target (source: our elaboration on Eurostat data).

In this energy transition process, the EU has promoted the formation of the "energy community" as a new entity that enables citizens (passive consumers) to become prosumers (active consumers and producers) and to play a proactive role in the deployment of renewable energy sources. Energy communities enable consumers to jointly pursue their individual and collective economic, environmental and social goals while contributing to the decarbonization of the EU energy system. Furthermore, energy communities, acting locally, can promote sustainable development in European cities and act as a driving force for the achievement of the Sustainable Development Goals (SDGs) 7 and 11 [2] and the EU climate and energy targets [25]. The gradual fragmentation of local energy production systems into a multiplicity of small, diffuse installations modifies the peculiar characteristics of heritage buildings and produces significant impacts on the urban and social system, which deserve to be investigated [26]. Indeed, local energy transition and climate adaptation actions have posed significant urban, social, economic and valuation challenges to cities that have required the development of adequate tools to guide and plan such transition within cities and their historical centres [27].

Although the energy sector is mainly controlled by state-granted or commercial companies, a recent study collected data in many European countries and provided the first quantification of the aggregate contribution to the European energy transition by citizen-led initiatives and projects in the energy sector [28,29]. Citizen-led initiatives are classified as formal or informal groups—e.g., energy cooperatives, eco-villages and also sustainable energy communities—which meet the following three criteria: citizen leadership; non-economic benefits, i.e., initiatives which do not pursue a profit; and activity in energy services provision, such as production and distribution of renewable energy as well as education activities for energy behavior change. Citizen-led initiatives are able to promote several types of projects, (e.g., to operate solar photovoltaic projects, to develop wind parks, to draft plans for developing smart villages, etc.) and involve a large number of people.

The highest concentrations of citizen-led energy initiatives, founded from 2000 to 2021, are in Germany and other northern European countries, e.g., the Netherlands, Denmark and Ireland (Figure 3). However, the corresponding number of people involved is widely variable depending on the type of promoted projects, so it can be large even for a small number of citizen-led energy initiatives, as in the case of Denmark, the Netherlands, Spain and Belgium [29]. Nevertheless, when the aggregate citizen-led energy initiatives were related to the population in 2021, the resulting indicator quantified citizens' commitment to directly participating in and promoting energy transition in each EU country (Figure 4). We can observe that this indicator has very low values in all EU member states—with the exception of Denmark, where it is equal to 525 people involved per 10,000 population—being below 50 for most states, e.g., 39 for Spain and 13 for Italy. This prevailing unwillingness of citizens to become involved could be a social factor that makes the spreading of energy communities slower and more difficult.

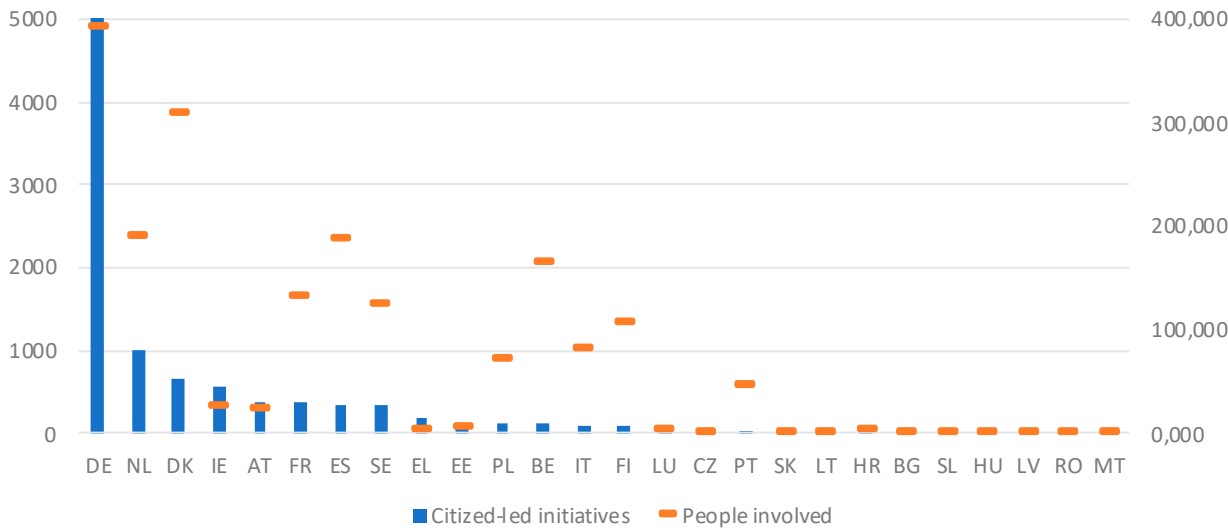

**Figure 3.** Citizen-led energy initiatives (left axis) and corresponding people involved (right axis) per EU countries (year 2000–2021) (source: our elaboration on Schwanitz et al., 2023 [29]).

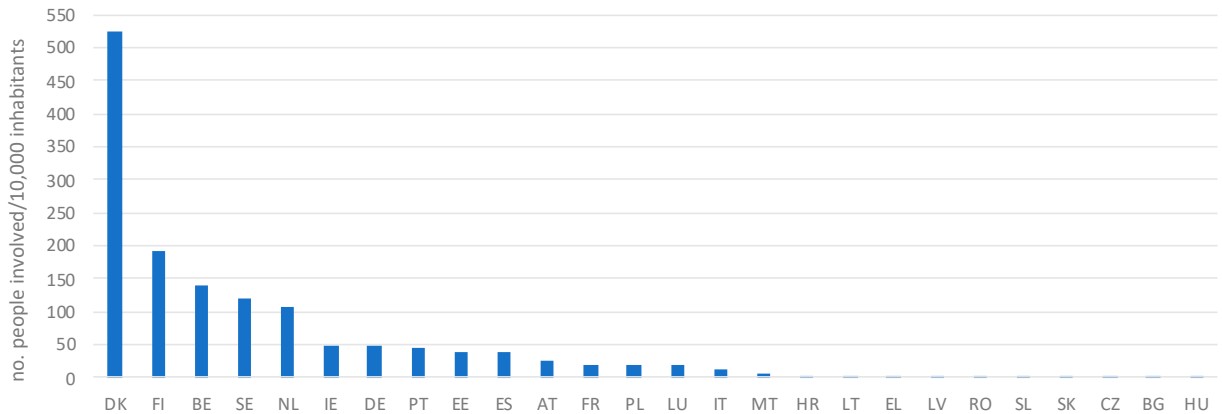

**Figure 4.** People involved in citizen-led energy initiatives per 10,000 inhabitants in EU countries (year 200–2021) (source: our elaboration on Eurostat and Schwanitz et al., 2023 [29]).

From a social perspective, however, energy communities can provide fair, direct and democratic access to energy resources, promote social cohesion and reduce energy poverty. Unfortunately, there is also the risk of accentuating social inequalities between states, regions and cities, as the degree of spread of energy communities is not equal among EU

member states due to peculiar economic and cultural factors, as well as significant differences in the transposition of European directives, such as the types of energy communities or the financial incentive instruments to support their formation.

Several studies have been conducted to investigate the role of citizens as actors in energy communities. Berrou and Soulier [30] developed a methodology based on the *Actor-Network Theory* model that helps to clarify the social dynamics of energy community formation; De Vidovich et al. investigated the organizational models of energy communities [31], while Musolino et al. [32] analyzed the influence of the local context on the composition and characteristics of actor-networks connected to energy communities (in particular, the differences between areas in northern and southern Italy). Other studies have investigated the social impact of energy communities in Europe [33] or the variation of social, economic and technical aspects of energy communities concerning the evolution of European directives on energy sharing [34].

Other fields of study have focused on the economic feasibility of forming energy communities, given the large number of actors involved and the need to allocate costs among community members to ensure investment recovery. Various business models (Energy Community Business Models—ECBMs) have been proposed to support the development of energy communities [35] or to guide decision-makers towards the types of energy communities that best meet specific policy objectives [36]. In parallel, models have been developed to maximize the economic feasibility of building energy communities through self-investment or third-party financing [37] and supporting the decision-making process of stakeholders involved in energy communities.

Several other studies have been carried out to compare policies to support self-consumption of energy in some EU countries [38]. Concerning Italian legislation, optimization models have been developed to guide the dimensioning and management of energy flows in energy communities [39] or to minimize operating costs and identify general guidelines for the optimal economic operation of energy communities [40]. Furthermore, studies have been conducted on particular cases of energy community formation, such as positive energy districts [41] or university campuses [42]. With regard to Spanish legislation, the size of optimal self-consumption installations [43] and the impact of incentive measures on the profitability of residential, commercial and industrial prosumers, as well as the variation of certain system conditions, were analyzed [44], including the sharing of domestic hot water production in energy communities [45]. The impacts of the implementation of the energy community of single-family building stock on a large scale have also been estimated in land and urban areas with certain peculiarities, such as rural areas [46].

Considering that the topic of energy communities is constantly and dynamically evolving, this study proposes a framework for analyzing and monitoring the effectiveness of regulatory and financial instruments that support the formation of energy communities. This framework refers to the four phases necessary to achieve the operation of an energy community and consists of three level of analysis, which are technical, financial and social. Energy communities are obviously promoted to achieve environmental goals (use of renewable energy sources and $CO_2$ reduction) but have the peculiarity of requiring the prerequisite of constituting a community of citizens. This makes it clear that energy communities also have important social goals to achieve in terms of social inclusion and combating energy poverty.

The proposed framework of analysis can be used for benchmarking, such as among EU states, and supports highlighting similarities and differences, bringing out best practices or any critical issues, as well as inefficiencies and inconsistencies to be corrected, in order to avoid creating or exacerbating territorial inequities at the local, national or European scale. The analysis of financial instruments is also related to four implementation phases of the energy community, whereas the economic feasibility of energy communities is assessed via economic indicators per type of financial instruments proposed by different nations.

The proposed framework is applied to two EU state members, Italy and Spain. These two states were chosen because they both have a near-average share of renewable energy sources in gross final energy consumption and, in addition, due to their climatic conditions and geographical location in southern Europe, have great potential for development, especially in photovoltaic and wind power generation. In contrast, the social habits of Italian and Spanish citizens to be directly involved in collective action in the energy sector has been very low so far, but it needs to be strengthened because it is the main factor on which energy communities are founded. The applied framework is a tool to point out the main critical issues affecting the effectiveness of energy transition strategies in Italy and Spain and assess the degree of economic feasibility for citizens involved in energy communities, based on the most recent legislative updates.

This paper is structured as follows: in the next section, the regulatory instruments on typologies of energy communities in the EU, Italy and Spain are presented. Section 3 illustrates the financial instruments to incentivize the formation of energy communities in Italy and Spain. The methods for testing the effectiveness of Italian and Spanish legislation on energy communities are described in Section 4. Section 5 discusses the qualitative and quantitative results of the comparison between Italian and Spanish energy communities. Finally, Section 6 presents conclusions and future research lines.

## 2. From Consumer to Prosumer: Typologies of Energy Communities

Energy communities have recently assumed a prominent role in energy transition policies, but significant differences between EU countries can be seen at the national, regional and local levels. Italy and Spain were chosen as southern European countries to evaluate the effectiveness of their strategies and the degree of cost efficiency for citizens. Indeed, Italy and Spain have a very low number of citizen-led energy actions compared to their population and only in the last two years has there been an upward trend in the spread of energy communities due to new regulations and incentives.

The study is organized in the following phases:

- Comparative analysis of the types of energy communities in the European Union, Italy and Spain through the review of current legislation;
- Comparative analysis of the financial instruments available for promoting the constitution of energy communities in Italy and Spain;
- Evaluation of the procedural, technical, economic and social critical issues related to energy communities.

### 2.1. European Union Guidelines on Energy Communities

According to the European Commission, energy communities are structured groups that can participate in any stage of the energy supply chain. Specifically, an energy community is a legal entity formed openly and voluntarily by members or shareholders—such as natural persons, companies, local authorities or public administrations—who choose to provide themselves with infrastructure for the production of energy from renewable sources through a model based on sharing, in accordance with the Renewable Energy Directive II (RED II Directive) [29] and Internal Markets Electricity Directive (IME Directive) [47].

Energy community members play an active role in the market by generating, consuming, sharing, storing or selling energy. As a result, consumers become prosumers unhinging the traditional supplier–consumer relationship and generating a profound transformation of the energy production model. The traditional centralized and hierarchical system, powered mainly by fossil fuels, is being replaced by a distributed and collaborative system, powered by renewable sources, which enables democratic access to autonomously managed energy. Indeed, energy communities are considered an essential tool to make energy transition fair and inclusive, allowing citizens to play a proactive role in their local energy system.

First, it is relevant to clarify the distinction between individual and collective self-consumption. Individual self-consumption relates to the single end-user, who produces

renewable energy for his own energy needs and can store or sell the surplus of energy produced to the market. Collective self-consumption, on the other hand, is based on a virtual model in which a group of users produces, stores and sells energy to other end-users who are either in the same building (pure collective self-consumption) or are geographically distant (community self-consumption).

According to EU directives, there can be different energy community models: the collective self-consumption and Renewable Energy Community (REC) (RED II Directive); or the Citizens Energy Community (CEC) (IME Directive).

The REC is an autonomous legal entity that invests in producing, selling and distributing energy (obtained from renewable sources) to those who are located in the vicinity of the production facilities. The main objective of a REC must be the provision of environmental, economic or social benefits to the community, its shareholders or members or the local areas in which it works, rather than making profits.

The CEC is a legal entity with no proximate link between community members and energy production facilities. In CECs, the rights of final customers are increased in terms of transparency of offers, contracts and bills; the possibility of setting up closed distribution systems has been regularized; and the liberalization of retail markets has been initiated, safeguarding the most vulnerable customers.

In summary, the main differences between Renewable Energy Community (REC) and Citizens Energy Community (CEC) are as follows:

- RECs are based on the principle of autonomy among the members, the constraint of proximity to power generation facilities and the management of different forms of energy (such as electricity, heat or gas) as long as they are generated from renewable sources;
- CECs have no proximity constraints and can manage electricity generated from both renewable and fossil sources.

These new forms of energy production and sharing support sustainable development at local level, as they offer numerous benefits to society:

- Environmental benefits, resulting from reduced emissions of $CO_2$ or other climate-changing gases through increased local and diffuse renewable energy sources;
- Social benefits, related to greater social cohesion resulting from citizen participation and cooperation and the mitigation of energy poverty;
- Economic benefits, related to a higher assurance of energy supply, less energy dispersion, shorter energy transport distances, bill savings from self-produced energy consumption, revenues from the sale of excess energy to other users and dissemination of both sharing economy and circular economy models.

While respecting the EU Directives, each EU member state has its own legislation (Table 1), characterized by different evolution in both content and timing of implementation, in which standards, constraints and characteristics of energy communities are specifically defined. The EU Directives regarding RECs and CECs, however, have provided a common and basic legal ground for all EU state members, while all national legislations enable citizens to shift towards renewable energy sources, adapting them to local economic and social conditions.

### 2.2. Tylogogies of Energy Communities in Italy

In 2019, Italy established two types of energy communities, which, in line with European directives, can be:

- Renewable Energy Self-Consumers;
- Renewable Energy Communities (RECs).

Renewable Energy Self-Consumers are end-users who, operating within the constraint of proximity, act collectively by producing renewable electricity for instantaneous and deferred collective self-consumption (through the use of storage batteries); therefore, they

may also store or sell self-produced renewable electricity, provided that these activities are not their main commercial or professional activity.

**Table 1.** Main legislation on energy communities in the European Union, Italy and Spain.

| EU | Italy | Spain |
|---|---|---|
| **Directive 2018/2001/EU** Renewable Energy Directive II (RED II) Formal and legal definition of collective self-consumption and Renewable Energy Community (REC) | **Decree-Law No. 162/2019** Definition of self-consumers and Renewable Energy Communities | **Royal Decree-Law 15/2018** Transposition of the RED II Directive |
| **Directive 2019/944/EU** Internal Markets Electricity Directive (IME)Formal and legal definition of Citizens Energy Community (CEC) | **ARERA Resolution 318/2020/R/eel** Rules for energy sharing | **Royal Decree 244/2019** Definition of energy communities with direct sales or a simplified net billing system |
| | **Ministerial Decree of 16/09/2020** Rules on the remuneration of self-production installations and collective energy consumption for energy communities | **Royal Decree 477/2021** Incentive programmes for self-producing energy installations |
| | **Legislative Decree 199/2021** Transposition of RED II and IME Directives | **Royal Decree 377/2022** Amendment to previous legislation |
| | **Draft Decree of the Minister of the Environment and Energy Security of February 2023** (not yet approved) Incentives for self-production installations and collective energy consumption for energy communities | |

RECs, on the other hand, are legal entities composed of sets of members (e.g., natural persons, local authorities, companies) located in proximity to renewable energy production facilities, who come together to produce and consume clean electricity voluntarily, in accordance with the principles of self-consumption and self-sufficiency. RECs must comply with the following constraints [48,49]:

- physical proximity for aggregations between members;
- maximum power limit for installations of 1 MW;
- use of the existing electricity grid for energy sharing (paying system charges).

Furthermore, the process of forming a REC requires several steps and requirements that include:

- the establishment of a legal and autonomous entity relating to its members (natural persons, companies or local public administrations);
- the adoption of a statute;
- the compliance with the rules of private law contracts.

From a technical point of view, there are no constraints either on the type of renewable source or on the area where the renewable energy facility is to be installed, although the facility, as well as having to be located close to consumers, must comply with the obligation to be connected to the low-voltage electricity grid through the same MV/LV transformer substation. The installation of a *smart meter*, to collect real-time data on production, self-consumption, energy transfer or withdrawal from the grid is also required. The facility, however, does not have to be owned by the community but can be made available by one of the members or an external party.

When production exceeds consumption, the energy surplus can be placed in storage systems for use when renewable sources are unavailable (e.g., during the night) or when peak demand exceeds the amount of energy available at that moment. Alternatively, excess energy can be sold to the grid.

*2.3. Typologies of Energy Communities in Spain*

Spain was the first European state to introduce the possibility of collective self-consumption [50] based on the European RED II Directive in 2018 and defined the rules for the setting up of energy communities [51]. The result was a radical change to the country's energy management, as previous legislation hindered energy self-consumption, and the possibility of sharing energy without the requirement of a private distribution network opened up.

Two types of installations are allowed for collective self-consumption of energy:

- installations with a simplified net billing system;
- installations with direct sales.

The first category includes small domestic installations or larger systems, with a maximum output of 100 kW, that supply electricity to services or industrial buildings. The second category includes installations with a power output of more than 100 kW that sell electricity to the market.

The legislation allows nearby consumers to share a single installation, as long as one of the following conditions is satisfied:

- the distance between the consumers' properties must be less than 500 m (or their register numbers must share their initial 14 digits);
- the consumers must be connected to the same low-voltage (LV) network.

In addition, the consumers of a shared system must determine how to distribute the self-produced energy among all members through fixed sharing ratios.

## 3. Financial Instruments Supporting Energy Communities

Each country may adopt a set of financial instruments to support the deployment of energy communities according to its energy policies and to implement national and local environmental sustainability and economic development objectives.

The establishment of energy communities creates social benefits related to participative forms of cooperation involving the local community and environmental benefits resulting from the production of energy using renewable sources that contribute to global sustainability. The energy community members may also receive economic benefits, although these may not be the principal purpose of forming such a community.

The main economic benefit is, of course, the reduction in energy bills due to the self-production of energy, which compensates for the initial investment (costs of the establishment of the energy community, the construction of the renewable energy installation, etc.).

*3.1. Financial Instruments to Support Energy Communities in Italy*

To improve the economic feasibility of investments in energy communities, in Italy there are the following set of financial instruments:

- Feed in premium—credit on self-consumption of energy;
- Tax incentive—tax deduction;
- Investment grant—direct subsidy.

To promote the use of storage systems and the convergence of energy production and consumption, the Energy Services Manager (Gestore dei Servizi Energetici—GSE) has set an incentive (feed-in-premium), differentiated by the typology of energy community, which rewards the self-production and consumption of energy from renewable energy installations over 20 years. In addition, energy communities receive tax deductions for some items in their energy bills, corresponding to the avoided transfer of self-consumed energy into the grid (Table 2) [52].

**Table 2.** Typologies of incentives for energy communities in Italy.

| Typology of Collective Consumption | Feed-In Premium EUR/MWh | Tax Incentives EUR/MWh |
|---|---|---|
| Renewable Energy Self-Consumer | 100 | 10 |
| Renewable Energy Community (REC) | 110 | 8 |

Investment grants and tax deductions for the implementation of renewable energy installations could be obtained through other regulations, such as the *Ecobonus* [53], but, in these cases, it is mandatory that the installation be part of an overall system of energy efficiency measures for the building.

Investment grants are provided specifically for energy communities in a draft Decree by the Ministry of Environment and Energy Security of February 2023, which, however, is awaiting verification by the European Commission. These incentives are financed by the Italian National Recovery and Resilience Plan (Piano Nazionale di Ripresa e Resilienza—PNRR) [54] and may be granted to municipalities with a population of fewer than 5000 to implement collective self-consumption systems and RECs from renewable energy sources, with a contribution of up to 40% of the investment cost. Depending on the power capacity of the installation, maximum allowable costs are established, ranging from 1500 EUR/kW to 1050 EUR/kW, respectively, for installations up to 20kW and for installations greater than 200 kW and up to 1000 kW. In addition, this proposed decree establishes new values for shared energy premium tariffs, which range from 100 EUR to 200 EUR/kW, depending on the power of the installation and, for photovoltaic installations, are increased to consider the different levels of insolation in the national territory—e.g., the premium tariff to installations located in northern Italian regions is increased by 10 EUR/kW to compensate for the penalty resulting from low insulation. Should this draft decree be approved, the incentives could support both territorial rebalancing policies and local energy transition.

Finally, according to the current law (Ministerial Decree of 16 September 2020) [55], each member of an energy community can periodically receive an amount corresponding to the division of economic benefits according to rules that each community freely establishes in a contract.

These economic benefits result from:

- sale of surplus self-generated energy;
- incentives on self-consumed energy;
- savings from physical self-consumption.

To supplement this national legislation, Italian regions have promulgated their own regulations and incentives to promote energy communities at the local level and to integrate them into territorial policies.

The framework of regulatory and financial instruments is highly differentiated between the Italian regions in terms of, for example, the definition of eligible expenses that constitute the base value for the calculation of the incentive or the role of local government agencies, which are sometimes the exclusive recipients of grants and, in other cases, receive an additional premium for their membership in an REC.

The cases of the regions of Sicilia and Campania (in southern Italy), Emilia–Romagna and Friuli–Venezia Giulia (in northern Italy) are offered as examples of the variety of local incentives.

The Sicilian Region published a call in 2022 to "promote self-consumption, maximize local energy consumption, and lower energy costs for citizens and businesses, also in anticipation of the centrality that such aggregated forms of self-consumption will assume in the concrete implementation of the ecological transition promoted and supported by the PNRR" [56].

This call grants incentives to the Sicilian municipalities that commit to establishing at least one Renewable Energy Community by assuming the role of promoter and being responsible for:

- identifying a minimum initial core of REC members (at least 10% of whom live in energy poverty, according to the parameters of the Italian Energy Poverty Observatory—OIPE) [57];
- dealing with the legal establishment of the REC;
- facilitating the formation of at least one facility by making a municipally owned area available to the energy community.

The municipal government therefore has three options: build a renewable energy system; fund a third party to build at least one renewable energy system; or aggregate those willing to make their energy system available to community.

These incentives are investment grants which are formed by a fixed and a variable portion based on the number of inhabitants of the municipality (Table 3). Eligible expenses to be refunded are exclusively the costs of the REC's technical–economic feasibility study, the administrative and legal costs for the constitution of the Legal Entity and the application for registration of the Energy Community with the GSE. Thus, there are no incentives for the costs of implementation of renewable energy installations.

**Table 3.** Distribution of investment grants to Sicilian municipalities (our elaboration on data from Sicilian Region, 2022).

| Inhabitants No. | Fixed Grant EUR | Variable Grant | | |
|---|---|---|---|---|
| | | EUR/Residents | Minimum EUR | Maximum EUR |
| No. ≤ 5000 | 9500 | 0.80 | — | 4000 |
| 5000 < No. ≤ 10,000 | 9500 | 0.40 | 2000.40 | 4000 |
| 10,000 < No. ≤ 50,000 | 9500 | 0.20 | 2000.20 | 10,000 |
| 50,000 < No. ≤ 100,000 | 9500 | 0.18 | 9000.18 | 18,000 |
| 100,000 < No. ≤ 200,000 | 9500 | 0.15 | 15,000.15 | 30,000 |
| 200,000 < No. ≤ 500,000 | 9500 | 0.10 | 20,000.10 | 50,000 |
| 50,000 < No. | 9500 | 0.08 | 40,000.08 | 63,398 |

The Campania Region provides investment grants exclusively to municipalities with fewer than 5000 inhabitants and that will take on the role of REC promoters. This grant has a maximum limit of 8000 EUR and can cover only expenses for the technical–economic feasibility study of the ERC, the administrative and legal costs for the constitution of the legal entity [58].

The Emilia–Romagna Region has provided also an investment grant for the costs of the REC's technical–economic feasibility study, administrative and legal expenses for the REC's establishment and the management of the project [59]. Grants are 80% of eligible expenses with a maximum limit of 50,000 EUR per REC. This grant can increase by 10% in particular cases of RECs, which include: RECs located in mountainous or inland areas as an incentive to counter depopulation; RECs that have among their members households living in energy poverty, social or public housing management or ownership entities; or local authorities that participate in the REC by providing areas or roofs of public buildings for the installations. Thus, unlike the Sicilian regulations, in this Italian region, grants can be offered to all entities that want to form an energy community, whereas the municipality's membership in the REC allows for an additional bonus.

The Friuli–Venezia Giulia Region provides grants exclusively to public entities; however, it extends the list of eligible expenses to include the costs of planning and implementation of photovoltaic systems, connection, storage systems, smart grids, REC technical–economic feasibility studies, administrative and legal expenses for the REC establishment and project management expenses [60]. The grant is 80% of the total eligible expenses with a maximum limit of 500,000 EUR per production facility.

Specifically, the subsidy for installation projects and construction costs is 30% of the investment cost of installations with a capacity of up to 500 kW and 45% of the additional cost for photovoltaic systems with a capacity of more than 500 kW—the additional cost is calculated compared with the average cost of a conventional energy installation (Table 4).

**Table 4.** Grants for the establishment of renewable energy communities in the Italian regions of Sicily, Campania, Emilia–Romagna, and Friuli–Venezia Giulia (our elaboration on data from Italian regions).

| Italian Region | Parties | Eligible Expenditures | Grants | | |
|---|---|---|---|---|---|
| | | | Amount | Min. EUR | Max. EUR |
| Sicily | Municipality with other parties | Technical and economic feasibility study, legal and administrative fees | Fixed + variable (depending on the No. of inhabitants) | 9500 | 63,398 |
| Campania | Municipality (if population <5000) | Technical and economic feasibility study, legal and administrative fees | Fixed | | 8000 |
| Emilia-Romagna | Public and/or private parties: | Technical and economic feasibility study, legal and administrative fees | 80% | | 50,000 |
| | • in mountain or inland areas<br>• with members in energy poverty<br>• owners/managers of public or social housing<br>• public entities that dispose of areas or roofs for installations | Technical and economic feasibility study, legal and administrative fees | 90% | | 50,000 |
| Friuli-Venezia Giulia | Public bodies | All types of works (from preliminary studies to construction and grid connection) | 80% | | 500,000 |

### 3.2. Financial Instruments to Support Energy Communities in Spain

In Spain, financial instruments to support energy communities aim to increase the economic viability of establishing energy communities through measures that promote both revenue generation, by regulating the conditions of sale of surplus energy, and cost reduction, by cancelling taxation on self-consumed energy. Thus, the financial instruments chosen are energy expenditure credits and tax exemptions. Non-repayable contributions are reserved for pilot projects or are provided by programs that incentivize self-consumption of renewable energy which many private and public entities, including RECs, can access.

Regarding power purchasing, during phases when the self-generation of energy is less than demand, prosumers buy imported electricity from the grid at market price (determined by a contract with a private electricity retailer or with a contracted company that charges a fixed tariff). However, during phases when PV energy production exceeds community demand, electricity exported to the grid is sold at an agreed price or at a price linked to the wholesale market price, depending on the type of collective self-consumption:

- In the simplified monthly net billing system, if the prosumer has a contract with a partnered retailer, the wholesale price of energy is charged, whereas in the case of a free-market electricity retailer, a price agreed between the parties is charged.
- In the direct-sale system, prosumers sell excess electricity to the grid like any other producer, i.e., at the wholesale price.

In both collective self-consumption systems, there is an exemption from both the grid access tariff (0.5 EUR/MWh) and the generation tariff (7%).

Some differences between the two systems concern the administrative procedure and revenue characteristics. The monthly net billing system has the advantage of simple

administrative and technical requirements. Revenues from surplus electricity transferred to the grid are discounted directly from the electricity bill with a monthly credit and are not subject to taxation since they are obtained in the form of savings, although they cannot be high due to the capacity limit of the installations.

The direct-sale system requires more complex administrative and technical procedures, but there are no limits to potential profits since there is no constraint on maximum installation capacity or monthly billing.

In 2021, six incentive programs were launched for the implementation of self-consumption installations with renewable energy sources, with or without storage systems in many sectors—the service sector, production sectors, residential sector, public administration and third sector (Programs 1, 2 and 4)—but also for the inclusion of storage systems in existing self-consumption installations (Programs 3 and 5) and finally for the implementation of renewable thermal energy installations in the residential sector (Program 6) (Real Decreto 477/2021) [61].

Different types of entities, including RECs and city energy communities, can benefit from these programs. Eligible expenses cover all works and facilities required for renewable energy production, e.g., cost of construction of structures, installations, monitoring systems, low-voltage and high-voltage electrical systems, edifications, demolition, the cost of project design and management, operating expenses, etc. Instead, the fees of formation of RECs are excluded.

The Spanish incentive system is very complex because it depends not only on the type of program, but also on many other factors, including the basic costs of installation and generation of the systems, which are differentiated by type of renewable source, and size of company, with cost percentages ranging from 15% to 65%. Alternatively, the subsidy is established based on the type and power of the system, e.g., from 250 to 2250 EUR/kW, but differs according to the typology of the ultimate beneficiaries (e.g., natural persons, businesses or public housing ownership). Finally, there are the additional subsidies; for example, the incentive increases by 5% for municipalities with a population of up to 5000 or for non-urban municipalities with a population of up to 20,000, whose villages have a population of less than 5000.

Other subsidies for the formation of energy communities have been granted to pilot projects under the Recovery, Transformation and Resilience Plan (Plan de Recuperación, Transformación y Resiliencia—PRTR) via the CE-IMPLEMENTA program (December 2021) [62]. The program's goal is to promote social innovation and citizen participation in energy efficiency, renewable energy availability and electric mobility and identifies energy communities as a key player in the energy transition.

Pilot project subsidies cover 30% to 60% of eligible project costs through calls for proposals (4 calls have been made from December 2021 to February 2023). For renewable thermal energy and renewable electricity plants, the share is 60%, and eligible costs include design, administrative and legal expenses and, most importantly, installation costs. In the evaluation of the project proposal, particular importance is given to the proximity of REC members to the project location, innovative features and the social impact of the projects, especially regarding the presence of REC members with low incomes or social vulnerabilities and the location of projects in municipalities at risk of depopulation in order to achieve territorial and social cohesion objectives [63].

## 4. Verification of the Effectiveness of Legislation on Energy Communities in Italy and Spain

In order to verify the effectiveness of energy transition strategies and the economic feasibility of investment in energy communities, a framework of analysis is applied to the Italian and Spanish regulations, administrative procedures and financial incentives. This framework consists of:

- a qualitative and comparative analysis of laws and rules regarding energy community from a technical, social and financial point of view;

- an evaluation of economic feasibility of energy communities through the calculation of economic performance indicators.

The comparative analysis first focuses on the characteristics of the various types of energy communities established by law, and second on four main phases of the entire procedure for making an energy community operational. The phases considered are the "Feasibility Study" and "Development and Implementation" initial phases and "Activation" and "Operation" advanced phases.

The economic indicators are calculated with respect to some hypothetical case studies of energy communities composed of a variable number of households and with different incentives. In particular, it was assumed that three case studies focus on energy communities in which the number of households and, consequently, the size of the buildings varies. In case studies 1, 2 and 3, the energy community consists of 48, 72 and 214 households, respectively, with three members per household, living in 8- or 12-floor buildings. The photovoltaic energy installation is 100 kW$_p$ in cases 1 and 2 and 200 kW$_p$ in case 3. The characteristics of the case studies are presented in Figure 5.

| Data | Units | Case 1 (48 households) | Case 2 (72 households) | Case 3 (216 households) |
|---|---|---|---|---|
| Buildings | no. | 1 | 1 | 3 |
| Building floors | no. | 8 | 12 | 12 |
| Dwellings per floor | no. | 6 | 6 | 6 |
| Dwelling area | m$^2$ | 80 | 80 | 80 |
| Total dwellings | no. | 48 | 72 | 216 |
| Household members | no. | 3 | 3 | 3 |
| Installation power | kWh | 100 | 100 | 200 |
| Photovoltaic panel surface area | m$^2$ | 670 | 670 | 1340 |

**Figure 5.** Characteristics of the case studies 1, 2 and 3.

All case studies are analyzed against four different scenarios:

- Scenario REC_IT. In this scenario, the incentives available in Italy for Renewable Energy Community are considered;
- Scenario RESC_IT. In this scenario, the incentives available in Italy for Renewable Energy Self-Consumers are included;
- Scenario ISC_IT. This scenario consists of individual self-consumption in Italy, and no incentives are available;
- Scenario PR6_ES. In this scenario, the incentives of Program 6 available in Spain for renewable energy installations in the residential sector are considered.

However, it is important to clarify that the main goal of this study is not to optimize the size of a PV system, but rather to evaluate, for the same PV system, the impact of different or absent incentives on the cost-effectiveness of investment in community energy.

The economic performance indicators applied to the case studies are those most commonly used in the energy field, e.g., Net Present Value (NPV), Internal Rate of Return (IRR) and Payback Period (PBP).

The Net Present Value is the discounted present value of cash flows resulting from the difference between inflows and outflows over a period of time. An investment (or project) is profitable when the NPV is positive; therefore, an investment with negative NPV values does not achieve economic feasibility for investors. The NPV formula is as follows:

$$NPV = \sum_{t=1}^{n} \frac{(R_t - C_t)}{(1+r)^t} - C_0 \tag{1}$$

where: $R_t$ is revenues for year $t$; $C_t$ is costs for year $t$; $r$ is the discount rate; $n$ is the period of analysis; $C_0$ is the initial investment.

The Internal Rate of Return is the rate that makes the NPV of a project equal to zero; consequently, an investment or project with an IRR greater than its cost of capital is profitable. The IRR formula is as follows:

$$\sum_{t=1}^{n} \frac{(R_t - C_t)}{(1 + IRR)^t} - C_0 = 0 \tag{2}$$

Both indicators, NPV and IRR, express the economic feasibility of a project from the investor's point of view.

Finally, the Payback Period represents the point in time when the flows generated by the investment cover the initial costs (Ci) that triggered it. The PBP formula is as follows:

$$PBP = \frac{C_i}{\sum_{t=0}^{n}(R_t - C_t)_0} \tag{3}$$

## 5. Discussing the Italy–Spain Comparison of Energy Communities

EU member states have contributed to creating the required political, legislative and economic conditions to achieve environmental sustainability goals. With specific reference to energy efficiency measures and energy production from renewable sources, Italy and Spain have promoted the dissemination of energy communities, recognizing their potential role in reducing the use of fossil energy sources and, consequently, greenhouse gas emissions.

Both countries allow individual self-consumption, so each consumer can potentially become a prosumer and provide the energy required for his own needs and, eventually, sell the surplus energy to the grid. Individual self-consumption is feasible mostly in low-density urban areas or rural areas where it is not viable to establish the energy communities due to technical and/or legal constraints (e.g., proximity or sharing the same transformer substation).

Collective self-consumption and energy communities are regulated differently in the two countries: Italy has made provision for both, while Spain has two different regulations for the sale of excess energy. Both self-consumption types and energy communities can provide environmental benefits from reduction of $CO_2$ emissions and use of renewable energy sources; social benefits related to active participation and cooperation of citizens and potential reduction of energy poverty; economic benefits in terms of savings on utility bills (from self-produced energy consumption) and from profits on the sale of surplus self-produced energy.

Table 5 shows that the profiles of energy communities differ mainly in terms of social benefits. In particular, even though energy poverty is mentioned as a social problem in many policy acts, current legislation does not provide for financial measures to support the expenses borne by those living in this condition, with the consequence of effectively excluding them from participation in the energy transition.

**Table 5.** Environmental and social benefits by typologies of energy communities in Italy and Spain.

| Country | Typology of Energy Community | Environmental Benefits | | Social Benefits | |
|---|---|---|---|---|---|
| | | Reduced $CO_2$ Emissions | Increased Use of Renewable Energy | Social Cohesion | Reduced Energy Poverty |
| Italy and Spain | Individual self-consumption | Low | Low | None | None |
| Italy | Collective self-consumption | Medium | Medium | High | Low |
| | Renewable Energy Communities | High | High | Medium | Medium |
| Spain | Collective self-consumption with direct sales | Medium | Medium | Medium | Low |
| | Collective self-consumption with simplified net billing | High | High | Medium | Low |

### 5.1. Critical Technical, Social and Financial Issues

To highlight some critical issues that need to be addressed by the respective countries in order to improve the promotion and deployment of energy communities at the local level, a framework analysis was applied to Italian and Spanish regulations on energy communities and energy sharing arrangements. This framework consists of three levels, which are technical, social and financial, and was detailed according to four main phases of formation of energy community:

- the preliminary feasibility and cost-effectiveness studies of the energy community project;
- the development and implementation phase of the energy community;
- the construction of the installations and the formal establishment of the energy community;
- the activation phase of the energy community, which includes the launching of the production and consumption of renewable energy and the granting of the incentives or tax benefits.

In some respects, mainly technical and social ones, the representation of critical issues that emerged in the two countries is quite similar (Table 6). From a technical point of view, many factors have a negative impact in both Italy and Spain, such as lack of specific expertise; unclear procedural processes; very lengthy administrative requirements; inadequate regulations; and slow procedures. In addition, these factors also make it slow and difficult to activate individual PV systems for self-consumption [38,64].

**Table 6.** Critical issues on energy communities by phase of procedure in Italy and Spain (source: own processing).

| Critical Issues | Initial Phase | | | | Advanced Phase | | | |
|---|---|---|---|---|---|---|---|---|
| | Feasibility Study | | Development and Implementation | | Activation | | Operation | |
| | Italy | Spain | Italy | Spain | Italy | Spain | Italy | Spain |
| Technical | Technical and specialized skills deficit | | Unclear and complex administrative procedures | | Long and complex administrative procedures | | Slow administrative procedures | |
| Social | Low knowledge and citizen involvement | | Difficulties in accessing technical and specialized knowledge | | Lack of motivation among community members | | Difficulties in shared energy management | |
| Financial | Difficulty in having investment capital | | High risk aversion of traditional lenders Difficulty in accessing financing | | High financial exposure | Average financial exposure | Slow provision of incentives | High incidence of fixed costs of energy bill |

From the social point of view, the problems common to Italy and Spain concern, first of all, the detection of a low level of knowledge about energy communities and their benefits [65], which affects the involvement of citizens and would require the launch of more efficient information and dissemination campaigns. Significant difficulties also relate to access to expert knowledge on procedures, timeframes and the amount of economic investment required to establish an energy community. Finally, difficulties have been encountered in shared energy management due to a regulatory framework that is still not fully adequate to address all the cases related to local energy networks.

Additionally, from a financial point of view, there are also similar critical issues in the two countries in both the low disposability of investment capital and access to financing, partly due to the absence of financial products specifically targeting small renewable energy investments. This leads to high initial producer exposure—mitigated in Spain by some incentives for installation construction costs—which is compounded by long waiting times for incentives or high fixed charges, which keep the perceived risk level of the investment high.

Italy and Spain, on the other hand, followed an opposite approach concerning the choice of the type of grants and incentives. Italy focused on the typology of feed-in-premium and tax incentives, while Spain provided investment grants. Table 7 shows how

these two approaches correspond to the provision of measures and financial incentives in the different stages of the realization of energy communities.

**Table 7.** Measures and incentives for energy communities by phase in Italy and Spain (source: own processing).

| Measures and incentives–Initial Phase | | | | Measures and incentives–Advanced Phase | | | |
|---|---|---|---|---|---|---|---|
| Feasibility Study | | Development and Implementation | | Activation | | Operation | |
| Italy | Spain | Italy | Spain | Italy | Spain | Italy | Spain |
| Local grants for feasibility study costs (in some regions) | — | Local grants for administrative and legal expenses | Co-funding for project design costs | (Only in case of building renovations: investment grants and tax incentives from other financial instruments, e.g., Ecobonus), | Investment grants for construction of self-consumption installations with renewable energy sources or storage systems | Bill savings achieved from self-generation of energy / Gain from the sale of self-produced energy to the grid / GSE feed-in-premium on self-produced and consumed energy / Tax deduction on energy bill (corresponding to the avoided transfer of energy into the grid) | — / Tax deduction from payment of grid access tariff and generation tariff |

In Italy, incentives are mainly related to self-produced and self-consumed energy, with the aim of supporting the return on capital invested in the operation phase of the installations. Additional regional grants are available to supplement the costs of the initial stages—costs of feasibility studies and other costs preparatory to the development and implementation of the energy community. However, in most cases, these incentives can only be accessed by municipalities and public entities that assume the role of promoters of energy communities. In this regard, it is important not to underestimate the potential benefits obtainable from the presence of grants for the feasibility study stage. Previous studies suggest that financial incentives should be considered in both the early and later stages of a project's life cycle [66]. In fact, having incentives in the early stages to investigate the economic feasibility and technical viability of an energy community allows for the mitigation of higher risks in the later stages and, in general, is a crucial element in all projects that require the involvement of multiple stakeholders and citizens, especially if there are citizens in socially and economically vulnerable conditions among them.

In Spain, according to the results of some studies [38,44], the legislation before 2021 did not yet guarantee adequate economic–financial instruments to facilitate the dissemination of energy communities throughout the country, and the profitability of investments in self-consumption energy installations was lower than in other countries. Instead, the programs provided by Royal Decree 477/2021 have focused on investment grants, i.e., a partial funding of the costs of building energy communities, and have made it possible to address the multiple financing needs of energy production and self-consumption facilities in different, including residential (Program 6), sectors, thereby decreasing the financial exposure of energy community construction by citizens and attracting more private investors.

Table 8 summarizes the main advantages and disadvantages corresponding to the two energy self-consumption typologies in Italy and Spain.

Despite a low-complexity administrative procedure, there are more constraints for collective self-consumption in Italy and self-consumption with simplified net billing in Spain than for the other two typologies of energy communities. These constraints mainly concern the proximity to power generation facilities in the first case, and the maximum power output of the installation in the second case, which consequently limits the production of photovoltaic energy and the potential profits from the sale of excess energy to the grid. In contrast, RECs in Italy and self-consumption communities with direct sales in Spain, allowing the participation of many members and large installations, ensure a wider coverage of energy consumption.

**Table 8.** Advantages and disadvantages by type of energy community in Italy and Spain (source: own processing).

| Country | Type of Energy Community | Advantages | Disadvantages |
|---|---|---|---|
| Italy | Collective self-consumption | Higher tax incentives than those for REC | Mandatory connection to the same LV/MV transformation cabin (secondary cabin) |
| | | The constitution of a legal entity is not required | The energy produced must be shared in the same place where it is generated |
| | | Administrative procedure shorter than that for REC | |
| | REC | Higher feed-in premium than that for collective self-consumption | Mandatory constitution of a legal entity |
| | | Potential membership by numerous public and private parties (citizen, entrepreneurs, municipalities, etc.) | Longer and more complex administrative procedure |
| | | Possibility of connection to the same primary (HV transformation) cabin instead of secondary cabin | |
| Spain | Simplified net invoicing | Simplified administrative procedure | Constraints on installation capacity |
| | | Exemption from grid access charges for excess energy | Limited revenues |
| | | Revenues exempt from taxation | Monthly revenue billing |
| | Direct sales | No constraints on installation capacity | Complex administrative procedure |
| | | No constraints on profit level | Payment of grid access tariff for excess electricity |
| | | No monthly revenue limit | Payment of energy generation tax |
| | | Exemption from grid access cost for excess energy | |

From an environmental sustainability point of view, all four types of energy communities help to address local energy supply problems but can cause visual impacts that alter the urban landscape if PV systems are not well integrated into buildings. However, the first two systems have a low visual impact as the energy installations should be mainly domestic and located on the roof of buildings, while the other two types of energy communities (CERs in Italy and direct sale in Spain), due to their large size, could generate a positive and significant impact on the reinforcement of renewable energy networks, but also a potential and strongly negative impact on the urban landscape, so their location in the urban landscape and their design must take into account the protection of architectural and identity values.

### 5.2. Variability in Economic Feasibility

The economic feasibility of the formation of energy communities in all four scenarios of the case studies 1, 2 and 3 (see Section 4) were evaluated on the basis of technical (e.g., energy produced, energy shared, etc.) and economic (e.g., capital invested, incentive on shared energy, etc.) data (Figure 6).

| Data | Unit | Case 1 - Scenarios | | | | Case 2 - Scenarios | | | | Case 3 - Scenarios | | | |
|---|---|---|---|---|---|---|---|---|---|---|---|---|---|
| | | REC_IT | RESC_IT | ISC_IT | PR6_ES | REC_IT | RESC_IT | ISC_IT | PR6_ES | REC_IT | RESC_IT | ISC_IT | PR6_ES |
| Photovoltaic system cost | €/kW$_p$ | 1500 | 1500 | 1500 | 1068 | 1500 | 1500 | 1500 | 852 | 1500 | 1500 | 1500 | 900 |
| Discount rate | % | 4 | 4 | 4 | 4 | 4 | 4 | 4 | 4 | 4 | 4 | 4 | 4 |
| Total energy consumption | kWh/year | 132,300 | 132,300 | 132,300 | 132,300 | 197,100 | 197,100 | 197,100 | 197,100 | 585,900 | 585,900 | 585,900 | 585,900 |
| Energy produced | kWh/year | 137,773 | 137,773 | 137,773 | 137,773 | 137,773 | 137,773 | 137,773 | 137,773 | 275,546 | 275,546 | 275,546 | 275,546 |
| Shared energy | kWh/year | 49,571 | 49,571 | 49,571 | 49,571 | 66,043 | 66,043 | 66,043 | 66,043 | 166,291 | 166,291 | 166,291 | 166,291 |
| Surplus energy | kWh/year | 86,806 | 86,806 | 86,806 | 86,806 | 70,335 | 70,335 | 70,335 | 70,335 | 107,859 | 107,859 | 107,859 | 107,859 |
| Energy self-sufficiency index | % | 38.52 | 38.52 | 38.52 | 38.52 | 34.22 | 34.22 | 34.22 | 34.22 | 28.62 | 28.62 | 28.62 | 28.62 |
| Capital invested | € | 150,000 | 150,000 | 150,000 | 106,800 | 150,000 | 150,000 | 150,000 | 85,200 | 300,000 | 300,000 | 300,000 | 180,000 |
| Installation cost per household | €/household | 3125 | 3125 | 3125 | 2225 | 2083 | 2083 | 2083 | 1183 | 1389 | 1389 | 1389 | 833 |
| Energy purchase price | €/kWh | 0.52 | 0.52 | 0.52 | 0.52 | 0.52 | 0.52 | 0.52 | 0.52 | 0.52 | 0.52 | 0.52 | 0.52 |
| Energy sale price | €/kWh | 0.05 | 0.05 | 0.05 | 0.05 | 0.05 | 0.05 | 0.05 | 0.05 | 0.05 | 0.05 | 0.05 | 0.05 |
| Revenues from energy fed into the grid | €/year | 6819 | 6819 | 6819 | 6819 | 6819 | 6819 | 6819 | 6819 | 13,708 | 13,708 | 13,708 | 13,708 |
| Savings from physical self-consumption | €/year | 726 | 726 | 726 | 726 | 726 | 726 | 726 | 726 | 726 | 726 | 726 | 726 |
| Operating costs | €/year | 400 | 400 | 400 | 400 | 400 | 400 | 400 | 400 | 600 | 600 | 600 | 600 |
| MISE incentive on shared energy | €/kWh/year | 0.11 | 0.10 | 0 | 0 | 0.11 | 0.10 | 0 | 0 | 0.11 | 0.10 | 0 | 0 |
| Refunds of tariff component | €/kWh/year | 0.008 | 0.01 | 0 | 0 | 0.008 | 0.01 | 0 | 0 | 0.008 | 0.01 | 0 | 0 |
| MISE incentive on shared energy | €/year | 5453 | 4957 | 0 | 0 | 7265 | 6604 | 0 | 0 | 18,292 | 16,629 | 0 | 0 |
| Refunds of tariff component | €/year | 407 | 407 | 0 | 0 | 543 | 543 | 0 | 0 | 1367 | 1367 | 0 | 0 |
| Avoided grid losses | €/year | 0 | 72 | 0 | 0 | 0 | 94 | 0 | 0 | 0 | 237 | 0 | 0 |

**Figure 6.** Technical and economic data of the case studies (source: own processing).

The resulting economic indicators NPV, IRR and PBP in Figure 7 were calculated by applying the formulas (1), (2) and (3) and using the RECON software (by ENEA) [67].

| Economic indicators | Unit | Case 1 - Scenarios | | | | Case 2 - Scenarios | | | | Case 3 - Scenarios | | | |
|---|---|---|---|---|---|---|---|---|---|---|---|---|---|
| | | REC_IT | RESC_IT | ISC_IT | PR6_ES | REC_IT | RESC_IT | ISC_IT | PR6_ES | REC_IT | RESC_IT | ISC_IT | PR6_ES |
| Net Present Value (NPV) at 20 years | € | 73,299 | 67,896 | -8721 | 34,479 | 99,934 | 92,772 | -8721 | 56,079 | 247,752 | 229,819 | -24,143 | 95,857 |
| Pay Back Period (PBP) | years | 13.2 | 13.6 | >20 | 15.6 | 11.6 | 12.0 | >20 | 12.7 | 9.8 | 10.34 | >20 | 13.6 |
| Internal Rate of Return (IRR) | % | 8.02 | 7.71 | 3.05 | 6.48 | 9.52 | 9.11 | 3.05 | 9.07 | 10.84 | 10.5 | 2.82 | 8.13 |

**Figure 7.** Economic indicators of the case studies (source: own processing).

The comparison of the economic indicators of the case studies (Figures 8 and 9) shows that the scenarios REC_IT and RESC_IT, corresponding to the two types of Italian energy communities, have the best economic performance in all the case studies. Given the same scenario, the change in NPV in the three case studies —for example from 73,999 to 247,752 EUR in case studies 1 and 3 respectively— depends on the higher cost efficiency and size of the PV system relative to the number of households involved and energy consumption. In contrast, given the same case study, the higher NPV value of the REC_IT scenarios depends on the feed-in tariff and tax deduction, which are higher than those of the other scenarios. Similar results are found for the indicators IRRs and PBPs. On the other hand, the NPVs of the scenario PR6_ES are also positive, although they are much lower than those of the scenario REC_IT, depending on the different type and size of financial instruments available in Spain.

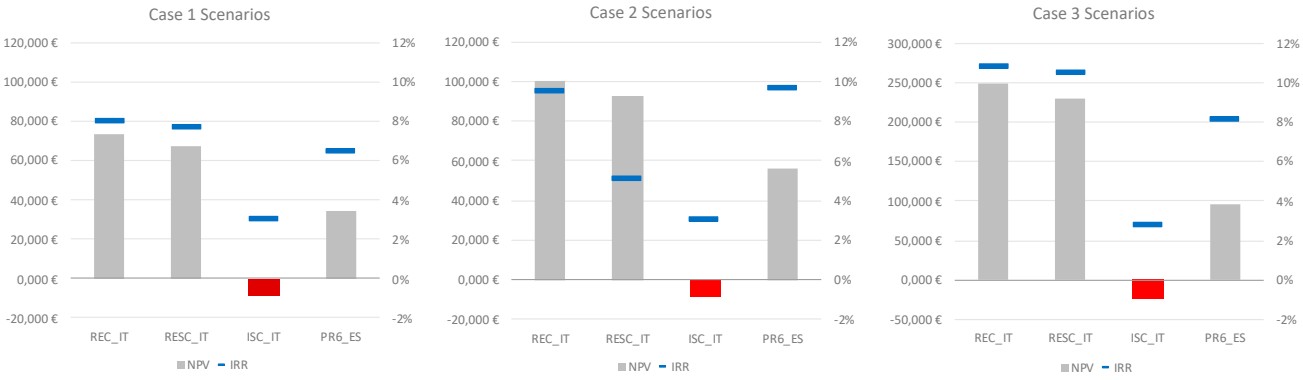

**Figure 8.** NPVs (left axis) and IRRs (right axis) of case studies 1, 2 and 3 (source: own processing).

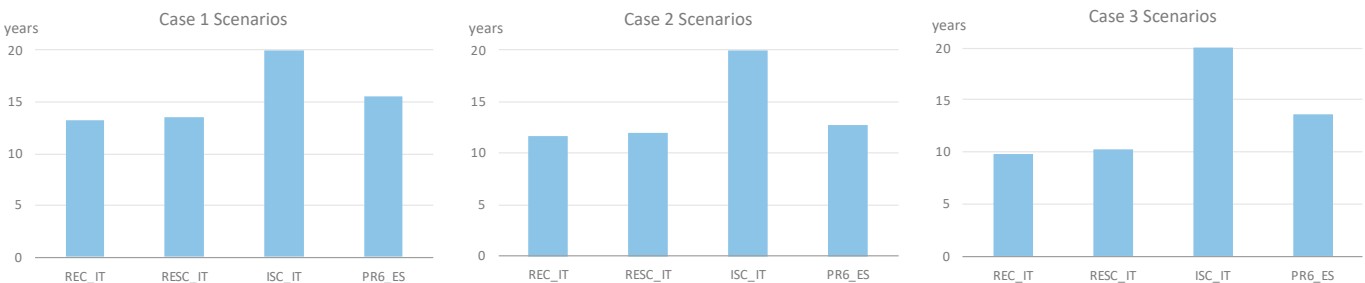

**Figure 9.** PBP of case studies 1, 2 and 3 (source: own processing).

It is worth noting that the three case studies always fail to meet economic feasibility without incentives; in fact, the NPVs of scenario ISC_IT are always negative and equal to -8721 EUR (case studies 1 and 2) or -24.143 EUR (case study 3), while the PBPs are always greater than 20 years. These results are indicative of the fact that such financial instruments are necessary to implement the energy transition in the current technical and economic conditions and to obtain the active involvement of citizens, as well as to increase the energy autonomy of cities.

*5.3. Policy Recommendations*

Energy communities are an instrument of environmental and energy policy and are based on the general principles of equity, energy autonomy and accessibility to energy sources. Their legal establishment is recent, so the transposition of general principles into norms and instruments is still being tested.

Assuming that there is no regulatory and financial system related to energy communities that can be adapted to every spatial, economic and social context; however, some policy recommendations can be pointed out based on the research and case study analysis presented in the previous sections.

*Regulatory flexibility/complexity*. While following EU directives, states and even regions can establish highly articulated regulations; of interest in this regard are the Programs for the implementation of self-consumption plants with renewable sources, which are differentiated by economic sector and type of energy (see Section 3.2). Obviously, more flexible regulations allow for better adaptation to the characteristics and needs of different sectors, but they also generate more bureaucratic complexity that can lengthen the time of bureaucratic processes.

*Systematic national/transnational monitoring*. In the absence of established procedures or practices, each country should conduct a periodic review of the consistency of regulation outcomes with overall energy transition goals; it would also be useful to document what is happening in other nations or regions. Monitoring data from the energy community should feed into an open-access database to facilitate the dissemination of best practices.

*Type and size of monetary incentives*. The choice of type and size of monetary incentives, while based on macroeconomic and microeconomic studies, must be subject to periodic revisions because changing initial technical, economic and social conditions may make them inefficient or ineffective. Of course, the combinations of incentive type/size are virtually infinite, but the cases already tested can provide some guidelines. For example, in the case studies presented, the scenario in which the energy community with the feed-in premium and tax incentives set by Italian law (REC_IT) is more convenient than that with the investment subsidy (PR6_ES) sets by Spanish law. From these results, it is necessary to explore the technical, financial and economic frontiers within which this convenience remains valid.

*Citizen involvement*. Energy communities are composed of citizens who need to be involved through local information and communication actions on energy transition and energy community issues. Implementing pilot projects, as has already been done for

NZEB buildings, provides tangible examples that can be shown to citizens in campaigns promoting energy communities and can facilitate their involvement.

*Fair and direct access to renewable energy resources.* Public authorities should pay more attention to social inclusion and energy poverty. Some social groups may have difficulty understanding the technical, procedural and economic aspects of energy community formation and affordability issues. These problems can be addressed by offering local counseling services, particularly in suburban neighborhoods, and by setting incentives specifically designed for them (possibly investment grants). In Italy, current legislation provides some incentives that offset the upfront costs (of design, legal fees, etc.) of energy communities, but this is not a sufficient measure to allow households in energy poverty to contribute to the overall expenses of the energy community and to prevent them from being excluded from the energy transition.

## 6. Conclusions

To strengthen ecological transition and reduce $CO_2$ emissions and energy consumption, the European Union has provided a legal groundwork for the formation of energy communities that transform citizens from passive consumers to prosumers and also enable citizens to play a proactive role in the diffusion of renewable energy sources in urban areas while gaining economic benefits. Furthermore, energy communities are also seen as a means of achieving a fair and inclusive energy transition.

As the field of energy communities is constantly and rapidly evolving and EU member states are providing very different actions to involve their citizens, this study proposed a framework of analysis of regulatory and financial instruments which consists of three levels and are related to each phase of the operating an energy community. In this framework, the effectiveness of the diffusion of energy communities in two European countries, Italy and Spain was analyzed, where the citizens' propensity for involvement in energy projects has been very weak but there is a great potential to be developed in photovoltaic and wind power generation. In particular, the regulatory and financial instruments adopted by the two countries were evaluated to determine whether they are adequate to promote the widespread formation of energy communities in urban areas and to meet economic feasibility of citizens involved in energy communities.

An analysis of the types of energy communities established in the EU, Italy and Spain has shown that they all produce environmental and social benefits, even if the issue of energy poverty is not yet effectively addressed. The comparative analysis of the financial instruments available to promote the establishment of energy communities, according to the most recent legislative updates, was the basis for the assessment of the economic feasibility of energy communities through the calculation of the most relevant economic performance indicators (Net Present Value, Internal Rate of Return and Payback Period) with respect to the different incentives currently available in Italy and Spain. According to the results reported in the previous section, a project with the same characteristics can reach different levels of economic feasibility depending on the type and size of incentives in place in the two countries, but above all, it is evident that the incentives are needed to make the formation of energy communities economically viable and, therefore, to succeed in involving citizens in the transition towards renewable energy sources.

The proposed framework of analysis made it possible to highlight the main procedural, technical, economic and social issues related to energy communities and which should be addressed by the two countries to implement energy communities more attractive. After all, the motivation for further efforts and investments to promote energy communities does not only concern the energy dimension but also the social and spatial dimension, since energy transition practices related to renewable energy sources can also become a tool for territorial policies to support social cohesion and inclusion and to increase the energy autonomy of cities.

Energy communities have only been legally established for a few years, so it may be useful to systematically monitor future changes in regulatory and financial instruments,

as well as technical and economic studies, so as to highlight critical issues to be corrected and best practices to be emulated. Further research could also extend the comparative study to other European states or different regions of the same nation to make the process of achieving environmental goals more efficient and avoid exacerbating territorial and social inequalities.

**Author Contributions:** Conceptualization, S.B.; methodology, S.B. and G.N.; formal analysis, S.B.; writing—review and editing, S.B. and G.N. All authors have read and agreed to the published version of the manuscript.

**Funding:** This research received no external funding.

**Conflicts of Interest:** The authors declare no conflict of interest.

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
