# Peer review of "Energy Communities in Urban Areas: Comparison of Energy Strategy and Economic Feasibility in Italy and Spain"

_land, doi:10.3390/land12071282_

Round 1

Reviewer 1 Report

The updated article appears remarkable and creative. Italian and Spanish urban energy communities' energy strategies and economic viability are compared. It is highlighting the cases of Spain and Italy. The current version can be much better if the following comments are considered in the revise version. The idea of energy and economic viability has been changed, although it is still unclear.  There must be a clear understanding of the current information due to the changed edition of the article.   

1.     In the abstract of the paper there is not showing the authentic concept of Italy and Spain, by comparing regulatory and financial instruments.

2.     No supporting information about the European Union's (EU) green transition is found in the introduction section. Updated literature is required.

3.     The information about "the EU has set targets for the reduction of CO2 emissions by 2030" was directly silenced in the second section of the introduction. Rephrase the entire sentence to include a CO2 notion.

4.     Make a clear fluency where the paragraph is linked to "In the energy transition process" in the introduction.

5.     There isn't a thorough explanation for figure 1-3.  Need more information on "Citizen-led energy actions per EU countries (year 2010-2021)" to make transparency in the given paragraph.

6.     Try to change the subtitles which can better reflect the message of the study theme and the appropriate context.

7.     What is the history of "Citizens Energy Communities CECs, and RECs," and how does producing energy help society?

8.     Table.1 does not provide adequate details regarding the primary EU legislation governing energy communities. Make sure to be concise and straightforward.

9.     In the case study and table 4, each case study is analyzed against four scenarios which is still unclear.

10.  Table no. 6 & 7 Critical issues on energy communities are not showing the clear message on the context of given case need modification scientifically.

11.  Modify Figure.  NPVs of case studies 1, 2 and 3. To describe a given case which is already discussed in the technical and economic data and economic indicators of the case studies.

12.  What is the significance of the research article after the whole paper reviewing there is no clear-cut importance given for the future studies try to add some about more research.

13.  The overall figures quality is not good. Please change the figures and upload in good quality.

Language is good.

Author Response

Dear reviewer, thank you very much for your work.

The point-by-point replies to the comments are contained in the attached pdf file.

Reviewer 2 Report

The study presents the energy communities in two selected countries (Italy and Sapin), focusing on the economic feasibility by comparing regulatory and financial instruments. It is much more than a simple description, some hypothetical cases are developed, calculating the most relevant cost-effectiveness indicators and comparing the existing regulations and procedures.

The structure is well-built and quite comprehensive, easy to read and follow.

I find the highest added value regarding the detailed presentation of the existing support systems, selection criteria, and weaknesses. It can be very useful for other Member States' policymakers to study. 

Because of that, I strongly recommend adding a Policy recommendation subchapter. It has to present the experienced best practices. With plain language: if any Member States (or another country in the world) want to build a complex system to support energy communities what the primary principles should be? What does an ideal system look like? What are the most severe problems that should be avoided? It doesn't have to be long, even 3/4-1 page is enough.

Other comments:

Fig.1: Please, sort the data in descending order.

Fig.2: Please, sort the data in descending order and remove the title from the chart.

Fig.3: Please, remove the title from the chart, and adjust the title of Figure 3. Now, it is the same as Fig.2 (but obviously it shows different data).

Author Response

(The authors gave the same response as above.)

Reviewer 3 Report

This study aims to assess the effectiveness of energy transition strategies and the economic feasibility for citizens of energy communities using renewable energy sources in two southern European countries, Italy and Spain. The topic is interesting and valuable. The paper is well organized.

I only have two suggestions:

1) In general, the research innovation is not clear enough. It would be better if the innovation and significance of this study could be clearly explained or emphasized in the text.

2) Why choose Spain and Italy? Please provide additional explanation on whether these two southern European countries are representative.

Author Response

(The authors gave the same response as above.)

Round 2

Reviewer 1 Report

I am satisfied with the modification. It can be published in current version.